# Exploring effects of severe mental illnesses on marriages: A qualitative study from Karachi, Pakistan

Sualeha Siddiq Shekhani[1¤a]*, Ahsan Mashhood[2], Durr-e-Sameen Hashmi[3¤b], Kiran Dossani-Lallany[4¤c], Nargis Asad[5], Murad Moosa Khan[6]

**1** Department of Psychiatry, Aga Khan University, Karachi, Pakistan, **2** Habib University, Karachi, Pakistan, **3** Department of Psychiatry, Aga Khan University, Karachi, Pakistan, **4** Department of Psychiatry, Aga Khan University, Karachi, Pakistan, **5** Department of Psychiatry, Aga Khan University, Karachi, Pakistan, **6** Department of Psychiatry and Brain & Mind Institute, Aga Khan University, Karachi, Pakistan

¤a Current Address: Centre of Biomedical Ethics and Culture, Sindh Institute of Urology and Transplantation, Karachi, Pakistan
¤b Current Address: Echoes Within (Private Practice), Agra, India
¤c Current Address: Toronto, Canada
* shekhanisualeha@gmail.com

## Abstract

Marriage is a central institution in South Asia, carrying deep social, cultural, and moral significance. In Pakistan, where psychiatric illness and divorce are both heavily stigmatized, the interaction between serious mental illness and marital life is complex and under-examined. Given the paucity of literature on this topic, our study explores how severe mental illnesses (Bipolar Disorder, Schizophrenia and Obsessive-Compulsive Disorder) affect marital quality, caregiving roles, and marriage outcomes within this sociocultural context. Using an exploratory qualitative design, we conducted in-depth interviews with 40+ participants recruited from a tertiary care hospital in Karachi. Participants were purposively sampled across 6 scenarios varying by disclosure timing and marriage outcome. Data were thematically analyzed using an interpretive approach grounded in family systems theory. Findings revealed that the presence of children, prior knowledge of the illness and the prevalent stigma of divorce were major influences in keeping the marriage intact. Spouse-participants reported significant caregiving strain, marked by role reversals, emotional exhaustion and health deterioration. Given the sociocultural and religious importance of marriage in Pakistan coupled with widespread stigma of mental illness, our findings call for culturally grounded mental health strategies that promote early disclosure, especially in the South Asian context.

**Data availability statement:** All data can be found in the manuscript and supporting information files.

**Funding:** The authors received no specific funding for this work.

**Competing interests:** The authors have declared that no competing interests exist.

## Introduction

Marriage is a central institution in South Asian societies and it plays an important role in social and family life [1]. The relationship between marital life and mental health is complex and bidirectional. Epidemiological evidence suggests that being married can have a protective effect against mental ill-health, whereas severe marital discord or breakdown is associated with heightened psychological distress [2]. Conversely, the onset or presence of a psychiatric illness can place considerable strain on a marriage, often leading to deterioration in relationship quality. Studies have found significantly higher rates of marital conflict, separation, and divorce among individuals with serious mental disorders compared to the general population [3–5]. Severe Mental Illnesses (SMIs), including bipolar disorder (BPD) and schizophrenia, have been particularly linked with marital difficulties demonstrating that mental health and marital stability are deeply related, each influencing the other in profound ways [2,6].

Furthermore, SMIs not only the individuals diagnosed but also their spouses and family systems. Family systems theory posits that family members are interconnected, such that when one family member suffers, for example from a mental illness, it) can disrupt the functioning of the entire family unit [7]. Empirical evidence supports the systems' perspective in the context of marriage and mental health. Research from Pakistan, for instance, reported that over 90% of patients receiving treatment for depression experienced significant marital dissatisfaction [8]. A quantitative methods study from Pakistan suffering from severe mental illness demonstrated a low quality of life among all adult patients [9]. Likewise in India, spouses of persons with serious psychiatric disorders exhibit markedly higher psychological distress and lower overall well-being compared to spouses in the general population [10]. Such findings further sediment the bidirectional spillover between individual psychopathology and couple dynamics: an untreated mental disorder can erode marital harmony and caregiver health, while a strained marriage may in turn exacerbate the course of illness [2].

In countries like Pakistan and India, being unmarried—especially for women—is often stigmatized and viewed as socially undesirable [11]. "Arranged" marriages (where the union is conducted primarily by individuals other than the couple themselves, such as parents or other kin members) are the norm in Pakistan. According to a Gallup Poll conducted in 2024, 81% of individuals in Pakistan report an arranged marriage although this is now changing in some urban settings [12]. This norm has two important implications for the nexus of mental health and marriage. First, the powerful societal pressure to marry often leads families to arrange marriages for individuals with mental illness due to the hope that wedlock might "solve" or mitigate their psychiatric problems [13]. Indeed, a pervasive misconception in South Asian societies is that marriage is a panacea for mental illness; as local mental health experts have noted, many believe that finding a spouse will cure conditions like depression or psychosis [14]. Second, despite the high value placed on marriage, mental illness itself remains a highly stigmatized condition in the community. Families often hide a member's psychiatric illness during the matchmaking process for fear of rejection by the prospective family [15,16]. Accordingly, stigma and lack of awareness thus create

a fraught environment in which psychiatric illness and marriage intersect, i.e., individuals may delay treatment to preserve marriageability, and those who are married may receive little community support, as mental health problems are expected to be borne silently within the family system [17].

Despite the considerable burden of mental illness in Pakistan and its evident repercussions on family life, there is paucity of research examining how SMIs affect marital relationships in the country. Much of the existing literature on mental health and marriage comes from either from Western settings or from India [18–20]. Scholarly attention from Pakistan to issues like marital adjustment, spousal roles, and the sociocultural challenges faced by couples dealing with SMIs in Pakistan has been limited. While there have been studies that investigate quality of life among caregivers of those suffering from SMIs, these have largely been quantitative surveys rather than those that attempt to provide a holistic account of the lived experiences of both spouses and patients [21,22].

Accordingly, this study draws on family systems theory to present Pakistan-based qualitative accounts that (1) explain why some SMI-affected marriages remain intact while others end, incorporating both spouses' and patients' perspectives; (2) analyze diagnostic disclosure and in-law dynamics; and (3) describe the caregiving burdens and associated gendered role reversals specific to Pakistani households. In doing so, this study aims to fill a critical gap in the literature by providing evidence that can guide interventions to address marriage-related expectations in persons with SMIs.

## Methods

We used an exploratory qualitative study design to explore the lived experiences of individuals married to those diagnosed with SMIs in the case of an intact marriage (henceforth referred to as spouse participants), and the patients themselves in the case of separation or divorce (henceforth referred to as patient-participants). Qualitative approaches are well-suited to unpack the complex interplay of cultural norms, stigma, and personal narratives in mental health [23]. By enabling participants to share their stories in depth, qualitative research can reveal contextualized insights such as how gender, family expectations religious beliefs mediate the impact of mental illness on marriage that might be overlooked by quantitative surveys. Given calls by experts for understanding mental disorders "in a cultural context" in Pakistan[14] we felt that a qualitative inquiry was suitable to explore this sensitive topic and generate rich data.

Ethical approval for the study was obtained from the Ethics Review Committee of the Aga Khan University (Ref no: 2522-Psy-ERC-13), Karachi, Pakistan. All participants provided written informed consent for participation. The study complies with ethical standards outlined in the Declaration of Helsinki.

### Conceptual framework

The study was informed by the interpretive social sciences position which seeks to study the meanings of people's experiences. These meanings can be varied and multiple therefore highlighting the complexity that is an inherent part of human life [24]. It was grounded in the family systems theory developed by Murray Bowen that views family members as interdependent. This means that a problem for any member of the family has an effect on all others [25]. This framework was apt for a study exploring marriage given that in Pakistan's largely collectivistic culture, the family plays a significant role in the institution of marriage.

### Setting

The study was conducted in the clinical settings of the Department of Psychiatry at the Aga Khan University and Hospital (AKUH), Karachi. AKUH is a fee-for-service, private teaching hospital established in 1985. Approximately 20,000 patients attend the psychiatry out-patient clinics annually, out of which about a quarter are those visiting the clinic for the first time. There is also an 18-bedded in-patient unit and with an average annual admittance of 450–500 patients.

Data was collected between 08/07/2013 and 26/05/2015.

PLOS **Global Public Health**

## Participants

This study included individuals diagnosed with severe mental illnesses in the case of divorce or separation, and spouses of those diagnosed with these conditions in the case of an intact marriage. Three mental illnesses including bipolar disorder, schizophrenia and severe forms of obsessive-compulsive disorder formed part of the inclusion criteria. Decision to include these illnesses was based on the observation of the psychiatry consultants that in most cases these result in chronic patterns causing major disruptions in personal and interpersonal aspects of marital life.

## Sampling technique

The study used a purposive theoretical sampling method to recruit participants with specific characteristics which were relevant to the objectives of the study, therefore enhancing the depth of the findings [26,27]. Given the variation in the marriage setting, the study was initially divided into four scenarios as seen in Table 1. However, during the process of data collection, the data collectors noticed variations in marriage setups and therefore the sampling strategy was modified with inclusion of two more cases (also noted in Table 1), remaining true to the iterative method characteristic of qualitative research. This variation was then relayed to the ERC in the bi-annual progress report.

Patients who lacked capacity (determined by the treating psychiatrist), patients with psychiatric conditions other than those mentioned above and those who refused consent were excluded from the study.

## Data collection tool

Demographic information was collected using a pre-designed questionnaire including name, age, duration of illness and marriage, the type of marriage (arranged by family, or by choice), employment status of the participants, family set-up (nuclear or joint family system), and number of children. A semi-structured interview guide (S1 Data) was prepared in line with the study's objectives informed by the existing literature on the subject. This contained specific probes that involved exploring the personal factors that led individuals to either stay or end the marriage. Socio-cultural factors such as the presence of family support and financial support, perspectives regarding divorce and presence of stigma were investigated. Questions related to the quality of marriage in terms of socialization outside as a couple/family, and the presence of children were also included. A section was also devoted to understanding the participants' subjective understanding of the disease, whether they believed it was a biological phenomenon or something rooted in the sociocultural such as black magic, or the evil eye etc. In addition, participants were asked about their outlook on the future and the coping mechanisms they used to deal with the illness.

The data collection tool was pilot tested with spouse-participants and patient-participants visiting the outpatient clinics, and modified accordingly.

## Data collection procedure

SSS and DSH, both with undergraduate and graduate education in psychology and mental health respectively, collected data for this study. The data collection took place between October 2013 and December 2015. Data collection continued

**Table 1. Sampling Frame—Division of Cases.**

| Cases | Participant | Diagnosis | Outcome of Marriage | Knowledge of Illness pre-marriage (Yes/No) |
|---|---|---|---|---|
| 1 | Spouse | Post-marriage | Intact | N/A |
| 2 | Patient | Post-marriage | Separation/Divorce | N/A |
| 3 | Spouse | Pre-marriage | Intact | Yes |
| 4 | Spouse | Pre-marriage | Intact | No |
| 5 | Patient | Pre-marriage | Separation/Divorce | Yes |
| 6 | Patient | Pre-marriage | Separation/ Divorce | No |

until recurrent findings such as presence of children, stigma of divorce and mental illness, along with significant caregiving stress were observed across participants. This was more specific for Case 1. For other cases, no more participants could be recruited.

The main primary investigators for the study (MMK and NA) trained SSS and DSH to collect data. After reviewing patients' files at both in-patient and out-patient departments, the data collectors approached those who met the inclusion criteria and requested them to participate in the study. Patients were interviewed while they were waiting to be seen in the clinic. In case they were called to see the psychiatrist, the interview was completed after the consultation. For in-patients, the process was easier as they were admitted to the ward. Therefore, we believe that the study did not impose any additional burden on participants with regards to time.

After obtaining consent, the interview took place, in a mixture of English and Urdu (the national language of Pakistan), in a private space ensuring privacy and confidentiality for participants. Each interview lasted for approximately 45–60 minutes. Patients were assured that they could refuse to answer any question that they wished to, and that they had the option to withdraw from the study at any time without any consequences on their treatment. Three potential patient-participants refused to participate in the study. Option for a referral to a mental healthcare professional was also provided to the spouse-participants.

Interviews were audio-recorded on a Smartphone and then later transferred to a Google Drive only accessible to the study team. Participants were also provided the option to not have their interviews recorded. Twenty-one participants refused the option of audio-recording. In that case, the data collectors took extensive notes during the interview.

Interviews were transcribed verbatim by SSS and DSH. Urdu quotes were translated to English by SS and verified by MMK. Identifying information was removed from the transcripts and notes for confidentiality purposes.

## Data analysis

Data was analyzed thematically using an inductive, interpretive approach grounded in Braun and Clarke's six-phase framework (See Fig 1) [28]. Transcripts were read multiple times for familiarization, followed by open coding conducted manually and independently by two authors (AM & SS).

Codes captured both descriptive and interpretive elements relevant to marriage, mental illness, stigma, and family dynamics. Coding was iterative, with collaboration among researchers to refine the final codebook. More specifically, themes were developed through constant comparison within and across the six case types. Guided by family systems theory, we paid particular attention to relational dynamics, caregiving roles, and the rippling effects of illness on marital stability. Divergent cases were retained to preserve complexity and nuance. In case of any disputes between the two coders, the mental health experts on the team (NA and MMK) provided feedback and resolved them.

## Rigor and trustworthiness

All researchers pursued reflexivity throughout the study particularly those involved in data collection (SSS & DH) and data analysis (AM & SSS). The research team was mindful of how researcher identities, particularly gender, can influence the process of data interpretation in research involving marriage and mental illness. To enhance analytical depth and reduce potential biases, the study benefited from a mixed-gender research team. SSS, a female scholar with a training in ethics and experience in conducting gender-related research, contributed a critical lens to understanding gendered dimensions of marital relationships. AM, a male researcher, brought prior experience in conducting sensitive research in mental health, which informed the analytical approach and interpretation of mental health narratives. DH, also female, contributed her training in psychology, which added depth while collecting data.

Rigor was also enhanced through (a) peer debriefing through the presence of two senior mental health professionals (MMK and NA) with wide experience in dealing with patients and their spouses, (b) audit trail memos, (c) triangulation

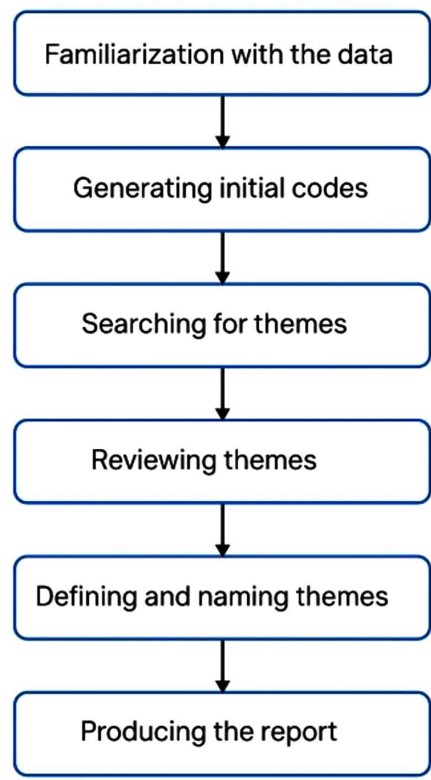

## Braun and Clarke's (2006) Coding Approach

Familiarization with the data

↓

Generating initial codes

↓

Searching for themes

↓

Reviewing themes

↓

Defining and naming themes

↓

Producing the report

**Fig 1. Approach for Data Analysis.**

across participant types (e.g., patients and spouses; different marriage outcomes [interact, separation/ divorce), and d) Generation of rich, thick descriptions regarding participants [29].

See Supplementary File 2 for adherence to COREQ checklist in S1 Checklist.

## Results

### Socio-demographic characteristics

Thirty-five spouses and 10 patients were interviewed for this study. The largest number of interviewees were in intact marriages (Case 1, n = 20; Case 3 n = 6; Case 4, n = 9). The remaining had either separated or divorced (Case 2, n = 5; Case 5, n = 3; Case 6, n = 2). Twenty-one participants were men, 24 were women. Twenty-nine participants were diagnosed with bipolar disorder, followed by schizophrenia (n = 13) and obsessive-compulsive disorder (n = 3). The age of the participants ranged from 21 years to 77 years. Thirty-three marriages were arranged by the family, whereas the rest were an outcome of a prior relationship, or "love" marriages, as they are colloquially referred to in Pakistani society (i.e., couples choosing their own marital partner). Twelve participants resided in joint family systems, whereas the rest had a nuclear setting. All but eight participants had children. The number of children ranged between 1 and 8.

Seventeen of those suffering from SMIs were hospitalized at the time of the interview, whereas the rest were recruited from out-patient clinics. Duration of marriage for intact marriages ranged from 6 months to forty-eight years. In the case of separation or divorce, the length of relationship ranged from 6 months to twenty-six years. See Table 2 for detailed break-down of sociodemographic details along with coded identifiers for each participant.

**Theme 1: Concealment, stigma and disclosure of illness.** There was a consistent lack of prior disclosure regarding one spouse's SMI/psychiatric diagnosis. Many participants either discovered the condition immediately after marriage: *"I found out on the second day of marriage… it should have been told to me before."* (C1-18) or only recognized its severity during a critical episode: *"I only realized it was a 'saya' [spiritual possession] turned into a psychiatric issue after eight years."* (C1-9). Spouse-participants who had not been informed about the mental illness of their prospective partner, prior to the marriage overall considered this as a form of deception (C4-6). They reported being *"shocked"* (C4-8), *"angry"* (C4-9) when they first found out about the illness, often made evident when it manifested itself in violent behavior. A wife stated that *"I should have picked it up because I always felt the hints were there. He used to get angry, beat himself up. Strange as it is, I never picked it up."* (C4-9) Partners also recounted that when they told their family members about the illness, *"they were angry, and the explanation we got from their side [in-laws], was rubbish, about why we were kept in the dark. That we didn't ask, so they didn't think of informing us."* (C4-9) On the other hand, one spouse-participant (husband) considered that this was part of *"..'kismet' (fate) that he would get married to someone with a mental illness."* (C4-4)

Patient-participants admitted that their families deliberately withheld information, fearing the proposal would be rejected or the marriage would not proceed otherwise. For one spouse-participant (C1-16), his wife's parents *"didn't want her in-laws to know she had bipolar… she seemed fine at the time of the wedding."* Participants described stigma around mental health as a primary motive for concealing the information. This pervasive withholding of information often under-mined a sense of trust once symptoms surfaced.

When questioned whether they would marry the individual if they possessed prior knowledge about the illness, major-ity of the spouse-participants replied in the negative, because: *"Why would anyone want their life to get ruined?"* (C4-4) and *"Why would I have put my life in such great difficulty?* (C4-8). According to another, *"even if you do get married, you shouldn't start a family or have a kid."* (C4-8). One husband, facing multiple issues in the marriage due to his wife's illness was indecisive, *"maybe, I am not sure. If you have any doubt, then you shouldn't marry."* (C4-1) On the other hand, another husband stated, *"I would have married her because I am used to dealing with tough situations in life."* (C4-7)

Even when disclosure occurred prior to marriage, interviews with both spouse-participants and patient-participants revealed that the quality of marriage was still negatively affected, regardless of the type of marriage (love or arranged). In arranged marriages, families often used the term 'depression' to inform the prospective family about the illness, *"My sister-in-law before marriage told me that he has depression. But I didn't know the extent of the 'depression.' I thought that sure, he may be depressed just like everyone is, living in Pakistan."* (C3-1) Another spouse-participant expressed that since she was quite young at the time of engagement to her now-husband, she could not understand the gravity and the problems that she would experience later on in her marriage. Eventually, *"the more I talked to him, I got interested in him. I was madly in love with him. I said to myself, so what if he has the illness? We will see. God is there to take care of me."* (C3-2) Inadequate understanding of the illness therefore led to marriage strain when spouses started living with their partners who had mental illness.

Knowing the spouse beforehand with a prior emotional bond did not serve as a protective factor in all marriages as evident in the case of C5-1, whose wife divorced him after 13 years of marriage. He shared, *"She took it as long as she could. And then she could not bear it anymore and then she left."* When questioned as to why she eventually left, he further shared, *"Not anything specific but a lot of incidents. Sporadic incidents. They kept happening. I started showing manic symptoms. My aggression increased, and her patience decreased. And that is a lethal combination. Between the-ory and reality, things are very different. She [the wife] knew the theory of it and when reality hit, she couldn't handle it."* (C5-1) This was evident in the case of a female patient-participant who believed she did not treat her husband well due to

**Table 2. Sociodemographic Details of Participants with Codes\*.**

| | Code | Case | Cate-gory | Gender | Diagnosis | Illness Duration | Age | Marriage duration | Type of marriage | Socio economic status | Family set-up | No of Cildren |
|---|---|---|---|---|---|---|---|---|---|---|---|---|
| 1 | C1-1 | 1 | Spouse | Male | Bipolar Disorder | 6 months | 50 | 18 | Arranged | Lower-middle class | Joint | 2 |
| 2 | C1-2 | 1 | Spouse | Female | Schizophrenia | 8 years | 35 | 8.5 | Arranged | Middle class | Nuclear | 1 |
| 3 | C1-3 | 1 | Spouse | Female | Bipolar Disorder | 5 years | 50 | 30 | Arranged | Lower-middle class | Nuclear | 4 |
| 4 | C1-4 | 1 | Spouse | Female | Bipolar Disorder | 2 years | 34 | 10 | Arranged | Lower middle class | Nuclear | 4 |
| 5 | C1-5 | 1 | Spouse | Female | Bipolar Disorder | 22 years | 57 | 35 | Arranged | Upper middle class | Nuclear | 3 |
| 6 | C1-6 | 1 | Spouse | Female | Bipolar Disorder | 2 years | 35 | 17 | Arranged | Middle class | Nuclear | 2 |
| 7 | C1-7 | 1 | Spouse | Female | Schizophrenia | 2 years | 50 | 3 years | Arranged | Middle class | Nuclear | 1 |
| 8 | C1-8 | 1 | Spouse | Male | Bipolar Disorder | 12 years | No response | 12 | Arranged | Middle class | Nuclear | 2 |
| 9 | C1-9 | 1 | Spouse | Female | Bipolar Disorder | 11 years | No response | 21 | Arranged | Middle class | Nuclear | 1 |
| 10 | C1-10 | 1 | Spouse | Female | Bipolar Disorder | 7 years | 39 | 14 | Arranged (second marriage for spouse) | Middle class | Nuclear | 3 |
| 11 | C1-11 | 1 | Spouse | Male | Bipolar Disorder | 10 years | 65 | 28 | Arranged | Upper middle class | Nuclear | 2 |
| 12 | C1-12 | 1 | Spouse | Male | Bipolar Disorder | 5 years | 64 | 39 | Arranged | Upper middle class | Nuclear | 2 |
| 13 | C1-13 | 1 | Spouse | Female | Bipolar Disorder | 3 months | 32 | 10 | Arranged | Middle class | Joint | 3 |
| 14 | C1-14 | 1 | Spouse | Male | Schizophrenia | 10 years | 48 | Unknown | Arranged | Upper middle class | Nuclear | 1 |
| 15 | C1-15 | 1 | Spouse | Male | OCD | 13 years | 58 | 32 years | Arranged | Middle class | Nuclear | 8 |
| 16 | C1-16 | 1 | Spouse | Male | Schizophrenia | 32 years | 57 | 33 years | Arranged | Middle class | Nuclear | 3 |
| 17 | C1-17 | 1 | Spouse | Male | Bipolar Disorder | 20 years | 50 years | 26 years | Arranged | Middle class | Joint | 2 |
| 18 | C1-18 | 1 | Spouse | Male | Schizophrenia | 6 months | 55 | 27 years | Arranged | Middle class | Nuclear | 4 |
| 19 | C1-19 | 1 | Spouse | Male | Bipolar Disorder | 1 week | 34 | 7 | Arranged | Middle class | Nuclear | 5 |
| 20 | C1-20 | 1 | Spouse | Female | Bipolar Disorder | 4 years | | 14 | Arranged | Middle class | Joint | 2 |
| 21 | C2-1 | 2 | Patient | Female | Schizophrenia | 17 years | 62 | 10 | Arranged | Lower-middle class | Nuclear | 2 |
| 22 | C2-2 | 2 | Patient | Female | Bipolar Disorder | 20 years | | 26 | Arranged | Middle class | Nuclear | 1 |
| 23 | C2-3 | 2 | Patient | Female | Bipolar Disorder | 9 | | 9 | Arranged | Middle class | Nuclear | 0 |
| 24 | C2-4 | 2 | Patient | Female | Schizophrenia | | | | Arranged | Upper middle class | Nuclear | 0 |
| 25 | C2-5 | 2 | Patient | Female | Schizophrenia | 3 years | | 2 | Arranged | | Nuclear | 1 |
| 26 | C3-1 | 3 | Spouse | Female | Bipolar Disorder | 14 years | 24 years | 7 | Arranged | Middle class | Joint | 2 |
| 27 | C3-2 | 3 | Spouse | Female | Bipolar Disorder | 8 years | 21 | 4 | Arranged | Middle class | Joint | |
| 28 | C3-3 | 3 | Spouse | Male | Obsessive-Compulsive Disorder | 25 years | 53 | 5 | Arranged | Middle class | Nuclear | 1 from previous marriage |
| 29 | C3-4 | 3 | Spouse | Female | Bipolar Disorder | 20 years | 40 | 20 years | Arranged | Lower middle class | Nuclear | 1 adopted child |

*(Continued)*

**Table 2.** (Continued)

| | Code | Case | Category | Gender | Diagnosis | Illness Duration | Age | Marriage duration | Type of marriage | Socio economic status | Family set-up | No of Cildren |
|---|---|---|---|---|---|---|---|---|---|---|---|---|
| 30 | C3-5 | 3 | Spouse | Male | Bipolar Disorder | 23 years | 29 years | 7 years | Arranged | Lower middle class | Joint | 2 |
| 31 | C3-6 | 3 | Spouse | Male | Bipolar Disorder | 5 years | 31 years | 2.5 years | Love | Lower middle class | Nuclear | 0 |
| 32 | C4-1 | 4 | Spouse | Male | Bipolar Disorder | 2 years | 25 years | 9 months | Arranged | Lower middle class | Joint | 0 |
| 33 | C4-2 | 4 | Spouse | Male | Bipolar Disorder | Since her chilhood | 32 years | 3 years | Arranged | Lower middle class | Joint | 1 |
| 34 | C4-3 | 4 | Spouse | Male | OCD | 7 years | 40 years | 6 years | Arranged | Middle class | Nuclear | 1 |
| 35 | C4-4 | 4 | Spouse | Male | Schizophrenia | Doesn't know | 31 years | 1 | Arranged | Lower middle class | Joint | 1 |
| 36 | C4-5 | 4 | Spouse | Male | Schizophrenia | 4 years | 30 years | 1.5 years | Arranged | Middle class | Joint | 1 |
| 37 | C4-6 | 4 | Spouse | Female | Bipolar Disorder | 19 years | 33 years | 11 years | Love | Middle class | Joint | 2 |
| 38 | C4-7 | 4 | Spouse | Male | Schizophrenia | Since her chilhood | 77 years | 48 years | Arranged | Upper middle class | Nuclear | 2 |
| 39 | C4-8 | 4 | Spouse | Female | Bipolar Disorder | 15 years | 28 years | 12 years | Arranged | Middle class | Nuclear | 3 |
| 40 | C4-9 | 4 | Spouse | Female | Bipolar Disorder | Before marriage -doesnot know | 28 years | 4 years | Arranged | Middle class | Nuclear | 1 |
| 41 | C5-1 | 5 | Patient | Male | Bipolar Disorder | 23 years | 41 years | 13 years | Love | Upper middle class | Nuclear | 2 |
| 42 | C5-2 | 5 | Patient | Female | Bipolar Disorder | 18 years | 36 years | 1 year 2 months | Arranged | Upper middle class | Nuclear | 0 |
| 43 | C5-3 | 5 | Patient | Female | Schizophrenia | 22 years | 41 years | 8 years | Love | Upper middle class | Nuclear | 0 |
| 44 | C6-1 | 6 | Patient | Male | Schizophrenia | 11 years | 35 years | 1.5 years | Arranged | Middle class | Nuclear | 0 |
| 45 | C6-2 | 6 | Patient | Female | Bipolar Disorder | 3 years | 26 years | 6 months | Arranged | Middle class | Nuclear | 0 |

*Participant demographic information is shown in Table 1. Coded identifiers are used in quotations to preserve anonymity. These details are included to offer a thick description of participants and support transferability.

her illness and expressed, *"I do owe him an apology. I do take the blame for the marriage ending. There is a lot of guilt in me and sadness in me that probably I did not behave the way I was meant to behave."* (C5-3).

This theme is well-supported: concealing a diagnosis appears to be the prevailing cultural norm, mentioned in over half the transcripts across diverse diagnoses, and it also reinforces that SMI threatens marital stability.

**Theme 2: Family dynamics and influence of in-laws.** The role of in-laws and extended family emerged as a crucial factor. Two sub-themes illustrate contrasting experiences of either support or blame.

In some cases, extended families provided robust emotional and financial support, thus facilitating marital stability. In C1-4, a spouse-participant stated, *"I live in a joint-family, my father-in-law is very nice… they handle most expenses and help with children."* According to another wife, *"I have ample amount of support. From my family, as well as my in-laws"* (C1-2). For one spouse-participant, whose husband was hospitalized for many weeks, believed that the support of her in-laws had made it easier to deal with the illness, *"Everyone is behind me. No one has left me alone at this hour. Whenever we need anyone from the family, they immediately come to our help"* (C1-13). This participant shared that she did not have to worry about household chores, or even taking care of her children when her husband was hospitalized, because

the in-laws were doing that, and she had to *"simply be there for my husband"* (C1-13) Therefore, she reported that she did not feel any *"Bojh" [burden]"* or any kind of *"worry."* For many wives, their relationship with their in-laws also improved once their husbands were diagnosed with the mental illness, *"my mother-in-law helps now after the illness. She takes care of the cooking and everything else."* (C1-20) In cases where wives were suffering from a mental illness, it was often her parents who stepped in to cover the cost of the treatment and extended moral support by taking care of the children (C1-1).

Conversely, many participants described in-law conflict marked by hostility and accusations. In one case, for instance, the wife, while facing physical abuse from her spouse with SMI, also struggled with difficulties in relationship with her in-laws, *"His brothers do not help us at all… in fact they blame me for his schizophrenia."* (C1-7) In fact many a times, spouse-participants stated that they were advised to divorce their mentally ill spouses by their parents, particularly mothers, *"when we first found about the illness, she [mother] said leave him, he will not get any better"* (C1-4). Parental concern stemmed from the ill spouse's impaired daily functioning, *"You cannot continue like this. He won't take care of the child. He's sleepy, doesn't go to office regularly, how will you deal with it?"* (C4-9) Families also suggested divorce because *"he was creating a lot of issues so my mother, my older sister and they all said that "leave him, leave him." I used to speak ill of my husband and that was my foolishness."* (C5-3)

While some parents did not directly mention divorce, they appeared to question the decision of spouse-participants, *"My parents said to me that all responsibilities are on you. What do you want? What should we do now? But I told them that I do not want to leave her. I have hope that she will get better."* (C4-4) However, as time progressed, family members also seemed to advise the healthy partners to stay in the marriage, because *"the illness was out of his control"* (C1-2) and that attempts should be made *"to make her better."* (C3-6)

We have moderate–high confidence that in-law relationships matter: nearly all participants noted extended family involvement, with effects ranging from supportive to negative. Pressures are gendered; daughters-in-law consistently reported higher (real and perceived) in-law demands.

**Theme 3: Daily burden and caregiving.** Regardless of the diagnosis, spouses often described intense caregiving demands. Two sub-themes emerged: (a) role reversal and (b) exhaustion/ burnout.

Nearly all the spouses in this study believed that due to the SMI of their spouse, they had to take additional responsibilities. For those who were married to someone with a more severe manifestation of the illness, majority of the tasks fell upon them which included tasks that are traditionally considered to be of the other spouse. For husbands, for example, while fulfilling the role of a breadwinner, they also had to take responsibilities of managing the house, and child-rearing, which they considered to be the domain of women.

Child-rearing, in particular, became the responsibility of the husband in case the wife was ill, *"I have brought up the children, I have bathed them, I have done everything"* (C1-16). Other tasks included *"making breakfast"* (C1-1) and *"giving them feeders"* (C1-19). Husbands also believed that the essential tasks of parenting such as teaching children the difference between right and wrong, and providing moral direction in life was an additional task they had taken upon themselves due to the vacuum provided by their wives. While some considered it their responsibility *"to support their wives"* (C1-1) others also considered it to be *'a problem"* (C1-16) and found it *"annoying"* when they had to do *"double work."* (C1-19) Husbands also believed that they had taken up those responsibilities which *"no one else would have taken"* (C4-3).

Such responsibilities often fell upon the husbands, in the case of intact marriages, because of the specific manifestation of the illness, for instance, in the case of one partner with a mental illness, homicidal intent towards the child became evident. The physician had then advised the husband to *"keep the child away from her"* and *"to take up responsibility of the child."* (C4-4). This also became pronounced because lack of support from other family members compounded the number of responsibilities that had to be undertaken by spouse-participant.

For healthy wives in the union, earning for the family became a responsibility although this was less evident, compared to men taking up the responsibilities of child-rearing. One spouse-participant for example, recounted that her husband

was unable to *"take care of the credit card bills, so I have to take care of it."* (C1-2). For one of the wives, coming home after a long day at work, and then cooking and looking after the child posed difficulties for her (C4-6). Wives also believed that because of their husband's illness, they had to take up responsibilities such as *"running to the bank and submitting children's school fees,"* (C4-8) something that they considered the responsibility of the father.

Many caregivers expressed anger, resentment, or depression arising from continuous strain. In C1-17 the wife said, "*I do vent out… I yell at him. But it falls on deaf ears… I have to keep going for our son.*" Sleep disruption, health problems (e.g., hormone imbalances, weight loss) were also common. Spouse-participants also reported *"I had to go for counseling, because it was getting very difficult for me to cope up especially with my son."* (C1-2) This was also because *"I feel I am forgetting things and my memory is getting weaker."* (C1-2) When questioned directly about their feelings due to the mental illness of their partner, spouse-participants reported feeling *"sad,"* *"crying all the time, because of the sadness"* (C1-9), *"feeling depressed and anxious"* (C1-9), *"feeling hopeless"* (C4-4) and *"stressed out"* (C1-2, C1-10). For one spouse-participant whose husband had displayed violent behavior, she reported feeling fearful all the time (C1-7). Others also believed that they had grown more *"sensitive"* due to their partner's illness, *"thinking about one thing constantly"* and *"often getting angry at very little things."* (C3-2) Amidst all the difficulties that spouse-participants were experiencing due to the mental illness of their partners, majority of them saw a *"bleak future"* (C1-2) especially because they did not know whether *"he will get better."* (C1-4)

On the basis of repeated detailed accounts, we have high confidence for burnout, role strain and role reversal across our interviews with different psychiatric conditions.

**Theme 4: Cultural & religious norms around marriage and divorce.** Our study revealed that one of the important reasons for intact marriages was the negative attitude towards divorce in the Pakistani society, common among both male and female participants, but more so for latter. They referred to strong cultural or religious disapproval of divorce. Even amid severe stress, couples felt pressured to stay married.

For a large number of spouse-participants, husbands and wives alike, divorce was not considered an option, since *"it is bad, I don't believe in it"* (C1-8) and a *"wrong act"* (C4-7). In fact, the aversion to divorce for many participants was such that *"separation should not even be thought about"* (C4-7, C1-18) and that every attempt should be made to *"compromise"* (C4-2) and *"for the couple to be patient with each other and work it out."* (C1-1) Interviewees also highlighted that problems are present in every marriage, and that *"One should strive to make the marriage work. Small things should not be blown out of proportion."* (C1-13)

While a few participants agreed that religion provided a recourse to divorce, keeping the marriage intact was still believed to be a priority because *"Allah has thought of something for you, that's why you are married. Yes, He has provided the option for divorce, but even then one should not think about it."* (C1-17). One spouse-participant also expressed that since divorce was so *"painful"* he did not consider it to be lawful within religion. On the other hand, a patient-participant who was divorced expressed, *"When religion allows it, why not?"* She had initially considered divorce to be *"taboo, and that it was better to die rather than to take divorce."* (C2-2)

Other spouse-participants also expressed the concept of "till death do us apart" and believed that they were meant to remain with their partner until one of them died (C1-18). Many others also considered the relationship between a husband and wife to be of utmost importance therefore, *"divorce is an extremely wrong act. You can ruin someone's life this way. What is there in life? For men, it is still easy to survive after divorce, but for a woman, it is not."* (C1-18) Another one expressed that while he may seek separation from his wife, *"she can live with her parents, and I can live on my own"* but *"I will never ever give divorce."* (C4-5)

A few participants highlighted that divorce should be undertaken only in certain circumstances, such as in the case of infidelity, *"I would have only considered divorce if she had illicit relations with any other man. This is not the case."* (C4-3) One spouse-participant also expressed that constant physical abuse may also lay the grounds for divorce, *"Well if there is a lot of beating, then a person should seek divorce."* (C1-7) It is interesting to note that for this particular participant, physical abuse was present in the early years of the marriage, but she chose to stay back in the marriage.

Despite the general negative attitude towards divorce, especially amongst those in intact marriages, a few spouse-participants shared that they had thought of divorce intermittently, *"When I am desperate, and I feel like I cannot carry on, I think of divorce, even after 14 years of relationship"* (C1-20). They reported thinking about this frequently *"whenever we have a fight"* (C1-2) and threatening their spouse that *"I am going to leave"* (C1-6), when the patient is hospitalized *"I often feel like running away. To be honest, I am more relaxed when he is not at home"* (C1-5). This also happened in instances when the patient was particularly violent, *"I feel like just going through with it [divorce] when she is acting this way."* (C4-2)

During times of violence exhibited by the mentally ill partners, spouses shared that they had left them for certain periods, *"I did run away once, but my child told me to come back"* (C1-9). One spouse-participant recounted an instance where he had gotten into a fight with his wife, and *"I broke a glass because I was so angry, and then I told her parents to take her away."* However, when his anger subsided, he went and *"brought her back"* (C4-4). Another wife shared how her husband had hit her multiple times, after *"which I left him. Then someone told me about this doctor, and take him for treatment, so I went back."* (C4-8) Thoughts regarding divorce also surfaced for spouses when they were feeling exhausted, *"There are times when I am very tired taking care of her, and then I think to myself, maybe I should end it, I am being honest here, but then I think no, let's fight this."* (C4-4)

Our participants also highlighted that patients also considered divorce as an option and it was not only a recourse for healthy partners. Due to the violent behavior of her husband and in-laws, one patient-participant chose to seek separation, because *"I cannot find love for him in my heart anymore."* (C6-2)

A handful of spouse-participants also expressed that while they wanted to leave their spouse, they felt trapped because *"He says he does not have anyone, even his brothers don't help him"* (C1-7) and *"if I leave him, his family will not help him. He will be sent to a mental asylum"* (C4-8) It appears as if they stayed in the marriage out of a sense of responsibility since there was no one else to take care of the patient, *"If I don't support him, then he will be completely lost."* (C1-7) Despite being unhappy in the marriage, she did not consider leaving her husband, suffering from schizophrenia, characterized by violent behavior, to be an option because *"How will I be at peace if I leave him [in this condition] when there is no one to take care of him?"* (C1-7) Her dilemma was further highlighted by the fact that she shared her feelings of fear of sharing a bedroom with him, tracing it back to how he had tried to strangle her while she was sleeping, and that staying married to him was not bringing her peace, *"I have a fire burning inside me. I am so afraid. How can I let go off this fear?"* (C1-7) Another spouse-participant also shared how his wife's parents were unable to take care of her, and therefore, he did not think he *"could leave her in this state. She will get worse there. They didn't get her the right treatment before [marriage]. How will they take care of her now?"* (C4-8)

Some participants stated that their families insisted that *"in our family, divorce is not an option"* (C1-18). Another participant (C1-13) fought until the very end, stating, *"Never… I am sure there are people more tolerant, so we should not divorce."* In rarer cases, second marriage (allowed in Islam) was suggested as a coping mechanism or alternative to divorce (C1-2), where the husband took a second wife due to first wife's illness; C1-18 *"I told her if I find a good girl, I might marry again… no wrongdoing in it."*

Our confidence in this theme is moderate because, while cultural norms appeared influential across interviewees cutting across class and socioeconomic status, the degree to which participants adhered to or resisted these norms differed based on individual beliefs and family expectations.

**Theme 5: Emotional and physical violence.** A subset of participants, mainly wives (both patient-participants and spouse-participants), experienced emotional abuse and, at times physical violence or intimidation from the spouse.

Husbands often used emotional abuse as a tactic particularly when they had discovered existence of mental illness after the marriage. According to one divorced patient-participant, *"He used to say bad stuff about my parents. He would say, 'you have given me your crazy daughter.' There was a lot of emotional torture. He never hit me though."* (C2-5). One spouse-participant reported hitting his wife during her manic phases, *"I get angry with her. Sometimes I raise my hand at her [in frustration]. I feel bad afterwards and then try to make amends once my anger has cooled down."* (C3-6). Some

participants recounted non-physical intimidation: locked rooms, taking away phones, or threatening a second marriage (multiple marriages are allowed in Islam for men) to manipulate. For example, in C1-20, a spouse-participant reported that her husband had *"threatened to lock me in the room"* while she was pregnant.

Spouse-participants particularly wives reported both emotional and physical violence during the active stages of the disease. One reported, *"His [ill husband]'s attitude changed a lot. There was a lot of swearing, a lot of cursing. To the extent that our children got disturbed."* (C1-4) Others, mainly women, however reported experiencing physical violence in varying degrees of intensity. Another one shared, *"Sometimes he has even mishandled me. But not too hard that I was hurt."* (C1-2). Yet another spouse-participant recounted that throughout the duration of her marriage to her husband diagnosed with bipolar disorder, there were several instances of violence directed towards her. One particular incident with dangerous levels of violence was in a particularly acute manic state, where her husband beat her with a belt and it escalated to the point that he tried to strangle her. She reported feeling *"shattered"* (C1-7) to the point that she kept crying for hours until finally she settled. Despite the real threat of violence and contemplation of divorce several times, she chose not to leave due to lack of support at her natal home.

We have moderate level of confidence here. While the data on emotional and physical violence is strong in those interviews where it appears, it is not consistent across all cases.

**Theme 6: Love, duty, and children.** For majority of spouse-participants, both male and female, the presence of children appeared to be a strong motivator in keeping the marriage together. In the case of intact marriages, thirty-three couples had children, whereas only two did not. In contrast, six out of ten patient-participants separated or divorced did not have any children.

When spouse-participants were asked the reasons for staying in the marriage despite the multiple difficulties that they faced in their marriage, responses such as *"because of kids"* (C1-3), *"and one has to think about kids"* (C3-5), and *"it is important for children to have both mother and father living together"* (C1-4) surfaced consistently. One stated that divorce should not be considered as an option, *"if there are children involved"* especially by *"women."* However, she believed that in case there were no children, then divorce could be a possibility (C4-8).

In instances where the quality of marriage was severely impacted due to the mental illness of their spouse, participants believed that this should not be *"taken out on the children"* (C4-5) due to the belief that *"children suffer in case of divorce"* and for the sake of the children, *"compromises should be made in a marriage."* (C1-9) Spouse-participants reported that they were trying to *"survive"* in the marriage because of children (C1-16). One wife recounted the instance of when she had *"left and went to my mother's place"* but *"came back after six months because I realized I had to continue. I have a 1.5 year old daughter"* despite repeated physical abuse. (C1-20)

Male spouses believed that even though their wives were suffering from mental illness, they were still *"someone's daughter"* particularly pronounced for those who also had daughters themselves (C3-5, C4-2). While many healthy husbands did not consider themselves close to their wives, they still viewed them *"the mother of their children."* (C1-16, C4-4) This was equally true in the case of healthy wives who believed, *"I can't take my daughters away from their father. I am nobody to do that, and I would never want it like that. What if my daughters ask me when they grow up, why did you leave him, he wasn't well, it wasn't his fault."* (C1-20) One healthy husband, in particular, reported that he had to stay back in the marriage because of the children since *"I don't have anyone else in the house to look after the children. No one will give love to my children. At least she [wife] will still give some love to them [since she is their mother]."* (C1-16).

Participants who were now divorced, and had children believed that the divorce had affected their children adversely, because they believed that *"the mental health of the children suffer."* (C2-5) One female patient-participant suffering from schizophrenia with one son recounted during the interview that she regrets that her husband divorced her because *"he [pointing to her child] is suffering"* since *"when he goes to his father to meet him, he feels it. What will he think about his father when he grows up?"* (C2-5)

A few spouse-participants also demonstrated some form of emotional attachment and a personal bond with their partners irrespective of the type of marriage (love or arranged), "*I am attached to my husband*" (C1-17). Wives in particular described a self-sacrificial or altruistic perspective: "*He's not normal, but that is acceptable to me*" (C1-16). One husband stated, "*I have suffered a lot because of her illness, and people have made fun of me, and no one has supported me. But I think about how she had supported when I was sick, and how she used to take care of me. Now when she is sick, I have to take care of her.*" (C1-16) Another husband reported how prior to the onset of the illness, his wife had supported him financially since she was a tailor "*who used to take care of her own expenses initially [in order to help me out].*" (C1-1) Another spouse-participant shared that when she was pregnant with their first child, her husband had taken care of her a lot, and after the mental illness became "*troublesome,*" she thought of leaving him, leading her to feel extremely "*guilty*" (C1-2).

In the absence of emotional attachment between the spouses, some spouse-participants perceived their decision to stay back in the marriage as a compromise they had made, stating, "*A relationship has been established. What can I do now?*" (C4-2) Others also conveyed a sense of resignation when they believed that this is how "*they will pass the rest of their days*" (C4-1). Another one after expressing his frustration in dealing with his wife stated that his attempt was to be patient, and leave his life up to God (C4-3).

We have high confidence in this theme: love, moral obligation, and especially children's futures recur in nearly all interviews that led to continued marriage. The data are rich, coherent, and transferable across BDP, schizophrenia, and OCD; dependability is shown by repeated reports of secrecy, caregiver burden, and in-law conflict.

Overall, we retain high confidence in the core themes: concealment, daily burden, cultural norms, and the centrality of children, but only moderate confidence about polygamy's prevalence and abuse intensity, which appear in limited subsets. Overall, the findings can be organized into push–pull dynamics in the context of marriage as shown in the table below (Table 3).

Across interviews, participants moved between these push-pull tensions, resulting in either a decision to remain in the marriage (often citing love, duty, or children) or a breakdown (when aggression and abuse, considerable stress, or stigma overshadowed any possible resolution).

In sum, the results reveal a complex interplay of social, cultural, and emotional factors guiding how spouses navigate marriage under the stress of mental illness. On the "push" side, undisclosed or poorly managed psychiatric conditions, critical in-law environments, role reversal, caregiver's burden and violence/aggression commonly pushed marriages toward separation. Conversely, "pull" factors like cultural aversion to divorce, presence of emotional bond, the presence of children, and supportive extended families often helped sustain marriages despite significant challenges. These findings underscore the necessity of disclosure prior to marriage, family-wide support, open communication which can help stigma reduction and prevent the cycle of hiding information and marital dissatisfaction frequently noted in the interviews.

**Table 3. Push and Pull factors.**

| Push Factors toward Divorce/Separation: | Pull Factors sustaining Marriage: |
|---|---|
| Non-disclosure of illness leading to betrayal or shock | Cultural/ religious aversion to divorce |
| Hostile in-law environment, stigma, blame | Family support (emotional, financial) |
| Emotional or physical abuse, controlling behaviors | Personal sense of love, loyalty, moral or religious duty |
| Escalating caretaker burden/ role strain | Concern for children's welfare |
| Severe untreated symptoms, psychosis, mania | Flexible coping strategies (e.g., separate household arrangements, increased caretaker involvement, continuing medication) |
| Financial or job instability leading to conflict | |

## Discussion

Our study presents the first in-depth qualitative analyses of how serious mental illnesses shape marital dynamics in Pakistan. Our findings, based on interviews with both patient-participants and spouse-participants, provide insights into the understudied but complex social, cultural and moral landscapes of what it means to navigate marriage as someone with serious mental illness, or be married to someone with one in the socio-cultural context of Pakistan.

Although the study data were collected over a decade ago, the participants' lived experiences and their willingness to share such intimate aspects of their marital journeys places a moral and ethical obligation on us to ensure their voices are heard [30]. Importantly, the social conditions in Pakistan has not undergone any significant transformation since the time of data collection. Mental illness and divorce remain highly stigmatized, and access to mental health services continues to be limited [31–33]. Thus, we argue that the relevance, depth, and contextual accuracy of our findings remain intact.

Our theme of concealment highlights how families withhold psychiatric histories, owing to culturally entrenched stigma of mental disorders. Such socio-cultural stigma is, of course, well documented in the case of Pakistan [34,35] and broadly, South Asia [36,37]. The practice of withholding such information, while ethically fraught, is socially understandable and common, even predictable or expected for a 'successful' marital match. This is in part because marriage in Pakistan is a union between two families (as exhibited in the high prevalence of arranged marriages) and not merely between two individuals. Accordingly, our data shows that this non-disclosure had enduring consequences that persisted in the marital life. For some, this also led to the ultimate dissolution of the marriage, a pattern which has been reported from India as well [14,38]. However, non-disclosure was rarely framed as individual deception alone, but rather as a socially patterned strategy by families, possibly to avoid social exclusion and marriage "market" rejection.

However, our second theme suggests that the role of family and in-laws cannot be placed into binaries of "always good/always bad." Families often served as buffers, offering financial, emotional, and logistical support, while others became sources of hostility, blame, and surveillance. Hence, unlike purely dyadic relations in the Global North, the *susraal* (in-laws family system) in the Pakistani context holds disproportionate power, and accordingly, it can be (1) either a social safety net (in the absence or formal state social support structures) or (2) a stressor, that adds to marital challenges and burdens associated with care [39,40]. Our study also illustrates the role of gender dynamics, given the patrilocal nature of the household in the Pakistani society, women were more likely to be subject to direct hostility and abuse from the in-laws, e.g., being blamed for the psychiatric illness of the spouse [31,41]. In cases where they were diagnosed with mental illnesses themselves, some of them also faced abuse from in-laws. The implication of this duality is that social interventions for Pakistani couples must draw on culturally responsive family systems theory.

Our third theme locates the impacts of role reversal and the subsequent caregiving burden that it placed upon spouse-participants, a common finding in literature from other parts of the world.[21] Both healthy husbands and wives described role reversals and increasingly intense caregiving demands. Men who typically occupied provider roles found themselves thrust into caregiving and child rearing; women, already burdened with emotional labor, often assumed financial responsibility as well. We therefore found that mental illness of a spouse had the effect, to varying degrees, of reconfiguring normative gender roles. This is an interesting observation given how gender roles within the Pakistani nuclear family are typically fixed, in part owing to a cultural expectation of marital life [42]. It is worth noting that this role reversal and associated burn out likely interacted with existing support by presence, or lack thereof, of family support.

Our fourth theme is in conversation with the broader societal perspective that sees divorce as the ultimate dishonour [43]. Despite extreme distress, respondents refused to consider divorce, for a variety of social, cultural and religious reasons, even when they saw a "bleak future" ahead. This is not surprising as ideas of marital suffering, sacrifice, and such "metaphorical martyrdom" in the name of family and framed as "compromise" are seen as positive qualities within the Pakistani context, especially for wives. There is also, as noted in theme 1, deep stigma, which renders divorced individuals less 'competitive' for future marriage, with second marriage preferred over divorce. This is largely resonant with existing evidence of the perceived isolation and social precariousness associated with the afterlife of divorce and may explain

why even survivors of violence in our study preferred staying in the marriage over the perceived indignity of exit [44]. Even for individuals that had sought divorce, the language they used was also religiously packaged including use phrases such as, *"allowed in religion."*

Our fifth theme centers on violence, both emotional and physical, more specifically directed towards women. This reflects the broader sociocultural dynamics in Pakistan [45,46]. A subset of wives in our study described emotionally abusive and at times physically violent behavior from their husbands, often during acute episodes of the illness. Based on our data, this violence ranged from swearing, intimidation and routine threats, to, as one participant described it, "being beaten with a belt and strangled during a manic episode." Interestingly, healthy partners themselves also admitted to emotionally degrading or physically harming their ill partner, particularly when the diagnosis had been concealed. One participant said he slapped his wife in frustration, later apologizing when his anger "cooled down." Such incidents were often normalized and justified as caregiving stress or marital exhaustion. Notably, even in cases of extreme violence, wives stayed citing stigma and lack of natal support. Hence violence rarely happened in a vacuum and was influenced by all previous themes.

Our final theme underscores the emotional and moral scaffolding that keeps many of these marriages intact: love, duty, and above all, children. For most healthy spouses, children were the most important 'protective factor' that appeared to sustain marriages. Participants repeatedly described how the presence of children made the idea of divorce unthinkable, even amidst mental illness-driven violence, neglect, or emotional disconnect. This reflects the broader moral script in Pakistan where children are considered to anchor marital stability, and where a "complete" family—mother and father under the same roof, is seen as essential to a child's well-being. Divorce, in contrast, was framed as a failure that children would eventually suffer for [33,47,48]. Even participants who had divorced expressed guilt about what the separation meant for their children's mental health, using language that focused on the future of children. In some instances, older children also played an active role by asking the parent to come back and stay together.

## Study implications

Our study provides valuable insights into the interplay of marriage and mental illness, underexplored through qualitative inquiries within the Pakistani context. The study findings are a microcosm of Pakistani society, where even in the absence of mental illness, sociocultural values such as stigma of divorce, and presence of children tend to hold marriages together. Mental illness in this context therefore adds another layer of complexity, but as our data illustrates, it does not necessarily result in separation/ divorce. However, lack of disclosure prior to the marriage leads to blame and fractured trust thus signaling the need for supported disclosure mediated through mental health professionals. Premarital counseling could be introduced within family planning frameworks, to discuss and explore disclosure, but also to equip couples with tools and expectations for supporting each other in the future.

## Study limitations

Despite the study providing a rich and layered account into the dynamics of mental illness shaping marriage, there are limitations to the study. The data was collected over a decade ago, and while we believe that the findings remain relevant given that such an extensive study utilizing qualitative methods has not been done within the Pakistani context, the findings should be interpreted keeping this time lag in mind. We also acknowledge that we have been able to capture only a subset of the Pakistani population drawn from one setting—recent information may be available about marital lives in the context of mental illness on other digital platforms particularly social media.

Theoretical sampling allowed us to gain rich variations and breadth in the type of cases but certain cases such as 3, 5 and 6 remained underexplored. While we appeared to reach data saturation in Case 1, this finding should also be viewed cautiously since we believe there is a high degree of probability that the mental illness may have been present before marriage and was diagnosed at a later point. Due to preponderance of Case 1 in our eventual sample size, there were

more spouse-participants as opposed to patient-participants. Therefore, our study findings highlight caregiving burden in marriages more rather than the lived experiences of being married for those with mental illnesses. Our study did not aim to generate generalizability therefore we cannot predict with certainty whether the patient's gender played a role in influencing the outcome of marriage. Quantitative data on marital outcomes among psychiatric populations in Pakistan may be useful in this regard. Moreover, considering the data was sourced from one tertiary care hospital in a cosmopolitan city, the findings may be different in other locations, particularly rural ones.

In addition, our cross-sectional study provided insights into the marital relationship at one point in time. However, longitudinal studies tracking couples with one partner suffering from a mental illness over time can help identify when relationships are most at risk and what predicts resilience.

Such data will be useful in developing marital counseling strategies as well as formal peer support groups, the latter being almost nonexistent in Pakistan. Ultimately, our findings suggest that SMIs act as a stress test for marriage. In fact, marriage in itself can be a potential risk factor for further stress for individual suffering from mental illness defying the more popular notion that 'marriage solves all problems' [14,49,50]. Within this context, community awareness is essential along with the important role of guided disclosures prior to marriage.

## Supporting Information

**S1 Data. Coded transcriptions of all cases.**
(ZIP)

**S1 Checklist. COREQ Checklist.**
(PDF)

## Acknowledgments

The authors would like to express gratitude to all the participants who volunteered to take part in the study. A special note to all the mental healthcare professionals at the Department of Psychiatry at AKU, Karachi who helped in accessing the participants.

## Author contributions

**Conceptualization:** Sualeha Siddiq Shekhani, Kiran Dossani-Lallany, Nargis Asad, Murad Moosa Khan.

**Data curation:** Sualeha Siddiq Shekhani, Durr-e-Sameen Hashmi, Nargis Asad, Murad Moosa Khan.

**Formal analysis:** Sualeha Siddiq Shekhani, Ahsan Mashhood, Durr-e-Sameen Hashmi, Nargis Asad, Murad Moosa Khan.

**Methodology:** Sualeha Siddiq Shekhani.

**Project administration:** Nargis Asad, Murad Moosa Khan.

**Supervision:** Nargis Asad, Murad Moosa Khan.

**Writing – original draft:** Sualeha Siddiq Shekhani, Ahsan Mashhood, Murad Moosa Khan.

**Writing – review & editing:** Sualeha Siddiq Shekhani, Ahsan Mashhood, Durr-e-Sameen Hashmi, Kiran Dossani-Lallany, Nargis Asad, Murad Moosa Khan.

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
