## [Decision Letter · Decision Letter 0]

26 Oct 2025

PGPH-D-25-02515

Exploring Effects of Severe Mental Disorders on Marriages: A Qualitative Study from Karachi, Pakistan

Dear Dr. Siddiq Shekhani,

Thank you for submitting your manuscript to PLOS Global Public Health. After careful consideration, we feel that it has merit but does not fully meet PLOS Global Public Health’s publication criteria as it currently stands. Therefore, we invite you to submit a revised version of the manuscript that addresses the points raised during the review process.

We look forward to receiving your revised manuscript.

Kind regards,

Abhijit Nadkarni

Academic Editor

Journal Requirements:

Additional Editor Comments (if provided):

Reviewers' comments:

Reviewer's Responses to Questions

**Comments to the Author**

1. Does this manuscript meet PLOS Global Public Health’s publication criteria?

Reviewer #1: Yes

Reviewer #2: Yes

2. Has the statistical analysis been performed appropriately and rigorously?

Reviewer #1: Yes

Reviewer #2: N/A

3. Have the authors made all data underlying the findings in their manuscript fully available (please refer to the Data Availability Statement at the start of the manuscript PDF file)?

Reviewer #1: Yes

Reviewer #2: Yes

4. Is the manuscript presented in an intelligible fashion and written in standard English?

Reviewer #1: Yes

Reviewer #2: Yes

Reviewer #1: This study investigates the effects of severe mental illnesses on marital quality, caregiving responsibilities, and marital outcomes in this particular sociocultural setting. My observations are given below:

(1) The study title is okay.

(2) The abstract needs minor revision. The abstract, which highlights the cultural dynamics of marriage and mental illness in Pakistan, is interesting and pertinent to the context. It is too detailed, though; readability and scholarly impact would be improved by more concisely expressing the research gaps, methodological support, and implications.

(3) The introduction also needs minor revision. With a strong emphasis on empirical data and pertinent South Asian context, the introduction offers a thorough summary of the connection between marriage and mental health. Its readability is hampered by its excessive length and some redundancy, though. Although there are many citations, some of the statements are still general and would be better served by critical synthesis as opposed to descriptive listing. Although the justification for concentrating on Pakistan is given, it could be strengthened by clearly contrasting it with previously published works of Indian and Western literature. The justification would be strengthened if the research gaps were more clearly stated. Improve the study rationale by including g the following studies in the introduction:

Tiwari, G. K., Singh, A., Choudhary, A., Shukla, A., Macorya, A. K., Pandey, A., & Singh, A. K. (2025). Forgiveness in Later Life: Attributes and Consequences for Older Adults in Indian Families. Marriage & Family Review, 61(7), 710–736. https://doi.org/10.1080/01494929.2025.2484381

Tiwari, G. K., Tiwari, R. P., Pandey, R., Ray, B., Dwivedi, A., Sharma, D. N., Singh, P., Tiwari, A. K., & Singh, A. K. (2024). Perceived Life Outcomes of Indian Children During the Early Phase of the COVID-19 Lockdown: The Protective Roles of Joint and Nuclear Families. Journal of Research and Health, 14(1), 43–54. https://doi.org/10.32598/JRH.14.1.1992.4

(4) The methods section also needs improvements. The methods section is thorough and shows a solid theoretical framework, careful application of qualitative design, and obvious ethical considerations. But the sampling procedure seems unduly complicated, and changing the criteria in the middle of the study might cause issues with transparency and consistency. For many participants, taking notes rather than recording audio could jeopardize the reliability and richness of the data. Although Braun and Clarke's framework was used, credibility would be increased with more information on inter-coder dependability and dispute resolution procedures. Although there is methodological rigor overall, it could use more explanation and clarity.

(5) The results section also needs revision. Although the results section provides a thorough and in-depth explanation of the sociodemographic traits of the participants as well as thematic findings, it is excessively descriptive and occasionally repetitious, which makes interpretation challenging. Large tables and lengthy narratives run the risk of overwhelming the reader. The section would benefit from a more succinct synthesis of patterns rather than a long list of cases, even though the use of participant quotes increases credibility. Stronger analytical depth and improved clarity could result from a more thorough integration of demographic data with thematic insights. It is also advised to make a clearer connection between the findings and the goals of the study.

(6) The discussion section also needs revision. The discussion section is thorough, organized, and provides profound cultural insights into how marriage dynamics and severe mental illness interact in Pakistan. Sometimes, though, it runs the risk of restating descriptive findings without critically integrating them with more comprehensive theoretical frameworks. Although the rationale for utilizing data that is ten years old is clarified, contextual validity remains a concern. Despite their richness, some themes might benefit from tighter synthesis to prevent repetition. Scholarly rigor would be improved by more interaction with cross-cultural literature and more precise explanation of the implications for policy and practice.

(7) The references are ok. Please correct it by following the journal’s guidelines. The tables and figures should be prepared following standard guidelines.

Reviewer #2: The study's aim and findings address important dimension of how mental illnesses influence marriages in Pakistan given the nature of patriarchal society and the pervasive stigma around mental illnesses, unmarried status and divorce. Although the use of decade old data is concerning, the authors have addressed and justified its relevance describing the lack of existing literature addressing this dimension, the continued stigma, and the lack of contextual changes with respect to norms and laws around marriages in Pakistan. However, given other kinds of rapid advancements including dissemination of information through digital platforms with implications for mental health awareness and the current economic problems in Pakistan, the use of a decade old dataset needs to be emphasised and acknowledged for where it might indicate gaps in creating knowledge representative of the present-day marital lives in Pakistan. Would also strongly advice revising the use of the term "broken marriage(s)" or marital outcome as intact or broken to move away from stigmatising separation or divorce. Instead, terms like marital status as separated or divorced can be used.

Title: Since the entire manuscript uses the term 'severe mental illnesses', hence, would suggest keeping it consistent and changing 'severe mental disorders' in title to 'severe mental illnesses'.

Introduction: Provides a concise brief introduction. Line 54 mentions 'studies' but only cites one study (3). Line 57-59 mentions 'one study' but cites two. Line 60 mentions 'existing literature' but cites the previously cited study only. Would suggest giving a richer introduction citing more existing studies especially from South Asia, not just repeat the same citation (2). If any statistics are available or qualitative research about the quality of life or problems faced in daily living by people with SMIs in Pakistan are available, would suggest including those here. in Line 63, would suggest revising the use of "disturbance in one member (such as mental illness)" since it situates and limits the mental illness within the individual ignoring its complex etiology. The use of word 'disturbance' can be replaced by the use of a more empathetic term like suffering.

Participants: The section mentions that "This study included individuals diagnosed with severe mental illnesses in the case of divorce or separation, and spouses of those diagnosed with these conditions in the case of an intact marriage." However, Case 4 in Table 1 includes spouses of patients in the case of "broken" marriage.

Setting: Check grammar in line 133 ("initial patients.-")

Data Collection Procedure: Line 176 needs clarification about how data saturation was defined for this study.

Table 2: under the column titled 'Marriage Duration", it needs to be clarified whether the duration is specified in months or years. The unit needs to be uniform across all rows.

Table 3: The use of the word 'universal' in Table 3 and in Results section to determine the trustworthiness of themes, seems problematic since qualitative research does not claim to produce universal findings anyways. Using the word 'consistency' for themes may be more appropriate when referring to the consistency of the themes within the dataset for the present study.

Although confidence levels are not required in qualitative research especially when ethical rigour and practices like reflexivity, use of audit trails, inter-coder relaiability have already been mentioned. CerQual is not necessary here since it is not a synthesis of evidence from different qualitative research. But it may be acceptable given the authors' justification for it when presenting each theme under the Results section.

Rigor and Trustworthiness: I appreciate the authors' attempts to transparently detail the ethical considerations and practice of rigour across the manuscript. Please check punctuations in line 225.

Results: In Line 494, would suggest to add "multiple marriages *for men* are allowed in Islam". In Line 620, specify the full form of 'HIC' (high income countries) since it is used here for the first time in the manuscript.

Discussion: Discussion section can be enriched with discussion of the present findings in light of more existing literature from South Asian countries or contrast with high-income counties or with findings from other nations with similar settings. Quantitative data from national-level surveys about family structure, age-wise and gender-wise distribution of SMIs, marital status of people with SMIs, etc. can also be discussed with reference to the present findings. Line 640 requires citation(s) of the "existing corpus of literature" mentioned.

Line 669 and 678- The phrases "children anchor marital stability" and "hold marriages together" can sound reductive and problematic when phrased as declarative statements.

Line 699- use of words like 'in fact' is problematic for something written without substantiating it with appropriate citations.

The last paragraphs mentioning the implications of the research should be put under a separate section for implications and future directions for use to separate them from the discussion of findings from the data.

I would suggest including a separate section on limitations to highlight the caution in using decade old data and data sourced from only one tertiary level hospital. The number of spouses (35) and participants (10; people with SMIs) interviewed also seems to be disproportionate and hence might highlight more caregiving burdens in marriages compared to the marital experiences of people with SMIs.

**Do you want your identity to be public for this peer review?** For information about this choice, including consent withdrawal, please see our Privacy Policy

Reviewer #1: **Yes: ** Gyanesh Kumar Tiwari

Reviewer #2: No

---

## [Decision Letter · Decision Letter 1]

2 Dec 2025

Exploring Effects of Severe Mental Disorders on Marriages: A Qualitative Study from Karachi, Pakistan

PGPH-D-25-02515R1

Dear Ms. Siddiq Shekhani,

We are pleased to inform you that your manuscript 'Exploring Effects of Severe Mental Disorders on Marriages: A Qualitative Study from Karachi, Pakistan' has been provisionally accepted for publication in PLOS Global Public Health.

Best regards,

Abhijit Nadkarni

Academic Editor

Reviewer Comments (if any, and for reference):

Reviewer's Responses to Questions

**Comments to the Author**

Reviewer #1: All comments have been addressed

publication criteria?

Reviewer #1: Yes

3. Has the statistical analysis been performed appropriately and rigorously?

Reviewer #1: Yes

4. Have the authors made all data underlying the findings in their manuscript fully available (please refer to the Data Availability Statement at the start of the manuscript PDF file)?

Reviewer #1: Yes

5. Is the manuscript presented in an intelligible fashion and written in standard English?

Reviewer #1: Yes

Reviewer #1: Well done. No further changes are required.

**Do you want your identity to be public for this peer review?** For information about this choice, including consent withdrawal, please see our Privacy Policy

Reviewer #1: **Yes: ** Gyanesh Kumar Tiwari
